# The Intestinal Microbiota in the Development of Chronic Liver Disease: Current Status

**DOI:** 10.3390/diagnostics13182960

**Published:** 2023-09-15

**Authors:** Josip Stojic, Michał Kukla, Ivica Grgurevic

**Affiliations:** 1Department of Gastroenterology, Hepatology and Clinical Nutrition, University Hospital Dubrava, 10000 Zagreb, Croatia; josip.stojic95@gmail.com; 2Department of Internal Medicine and Geriatrics, Faculty of Medicine, Jagellonian University Medical College, 31-688 Kraków, Poland; michal.kukla@uj.edu.pl; 3Department of Endoscopy, University Hospital, 30-688 Kraków, Poland; 4School of Medicine, University of Zagreb, 10000 Zagreb, Croatia; 5Faculty of Pharmacy and Biochemistry, University of Zagreb, 10000 Zagreb, Croatia

**Keywords:** intestinal microbiota, chronic liver disease, gut–liver axis, dysbiosis, gut permeability

## Abstract

Chronic liver disease (CLD) is a significant global health burden, leading to millions of deaths annually. The gut–liver axis plays a pivotal role in this context, allowing the transport of gut-derived products directly to the liver, as well as biological compounds from the liver to the intestine. The gut microbiota plays a significant role in maintaining the health of the digestive system. A change in gut microbiome composition as seen in dysbiosis is associated with immune dysregulation, altered energy and gut hormone regulation, and increased intestinal permeability, contributing to inflammatory mechanisms and damage to the liver, irrespective of the underlying etiology of CLD. The aim of this review is to present the current knowledge about the composition of the intestinal microbiome in healthy individuals and those with CLD, including the factors that affect this composition, the impact of the altered microbiome on the liver, and the mechanisms by which it occurs. Furthermore, this review analyzes the effects of gut microbiome modulation on the course of CLD, by using pharmacotherapy, nutrition, fecal microbiota transplantation, supplements, and probiotics. This review opens avenues for the translation of knowledge about gut–liver interplay into clinical practice as an additional tool to fight CLD and its complications.

## 1. Introduction

The intestinal microbiota is increasingly being discussed in the context of many diseases, and chronic liver disease (CLD) is no exception. According to Asrani et al., liver diseases cause approximately two million deaths per year worldwide, while the major complications of CLD—cirrhosis and hepatocellular cancer—account for 3.5% of all deaths worldwide [1]. The gut–liver axis indicates the bidirectional relationship between the gut and the liver. This mutual interaction is made possible by the portal vein, responsible for transporting 75% of the liver’s blood supply, including nutrients and microbial products, facilitating the direct transfer of gut-derived substances to the liver. Additionally, the liver reciprocates by releasing bile and antibodies back into the intestine through this pathway. This intricate relationship ensures seamless communication between the gut and liver, influencing various physiological processes in both organs [2]. The intestinal microbiota provides nutrient transformation through the fermentation of nondigestible substrates (e.g., dietary fibers). Also, it participates in the maturation of mucosal immunity, vitamin supply, and gut-to-brain communication [3,4]. The microbiota is constantly changing under the influence of many factors (e.g., nutrition, drugs, stress, immune system, host genetics, and diseases) which can cause natural variations in the microbiome composition but also progress to a state of dysbiosis. The microbiome can be described as a unique microbial community thriving in a specific and well-defined environment with distinct physiochemical properties. It encompasses not only the microorganisms present but also the diverse ecological niches they create through their activities. The microbiota, on the other hand, refers to the living microorganisms found in a particular environment, excluding nonliving entities like phages, viruses, plasmids, prions, viroids, and free DNA. The microbiota comprises microorganisms from different kingdoms, such as prokaryotes (bacteria, archaea) and eukaryotes (protozoa, fungi, and algae), along with various microbial structures, metabolites, mobile genetic elements like transposons, phages, and viruses, as well as relic DNA, all influenced by the environmental conditions of their habitat [5]. Dysbiosis represents a state of significant changes in microbiome structure, including a reduction in microbial diversity and a predominance of certain bacterial taxa as well as fungi, viruses, archaea, and helminths, causing an imbalance in the structure of the microbial community [3]. Dysbiosis is associated with immune dysregulation, altered energy and gut hormone regulation, and modified gut barrier function which increases intestinal permeability (IP) and allows the translocation of microbial products into the portal circulation triggering proinflammatory mechanisms [4,6,7,8].

The aim of this review is to present the current knowledge about the composition of the intestinal microbiota in healthy individuals and those with CLD, the factors influencing microbiome composition, the impact of the altered microbiome on the liver and the mechanisms by which it occurs, and the possibilities and benefits of diagnosing the microbiome composition in patients with different stages of CLD, as well as to analyze the possibility of modulating the composition of the intestinal microbiota as a preventive or therapeutic action in CLD.

## 2. Composition of the Human Gut Microbiome

### 2.1. The Gut Microbiome in Healthy Adults

The past decade has witnessed significant progress in culture-independent microbiologic technology, enabling a better understanding of the gut microbiome’s composition and diversity. However, defining a universal ‘normal’ microbiota remains challenging due to the considerable variations between individuals’ gut microbiomes. Nonetheless, in the feces of healthy individuals, the predominant bacterial phyla are *Bacteroidetes*, *Firmicutes*, *Proteobacteria*, and *Actinobacteria*, with *Bacteroidetes* and *Firmicutes* collectively making up around 90% of the gut microbiota [9,10]. The intestinal lumen harbors an abundant population of microorganisms, totaling trillions, encompassing over 1000 diverse species comprising bacteria, protozoa, archaea, fungi, and viruses [6]. Within the human intestine exists a vast gene pool of over 10^10^ microorganisms, collectively known as the human gut microbiome [11]. Tap et al. reported that the gut bacterial composition primarily comprises *Firmicutes* (79.4%), *Bacteroidetes* (16.9%), *Actinobacteria* (2.5%), *Proteobacteria* (1%), and *Verrumicrobia* (0.1%), identified through the utilization of 16S rRNA sequencing techniques [12]. Furthermore, enterotypes in the human gut microbiome are used for classifying individuals based on the gut microbiome and its varieties of microbial taxa [13,14]. Enterotypes are clusters of bacterial communities in the gut, representing symbiotic balanced states and showing different responses to various factors (e.g., gender, age, medications, diet, geographical distance). Originally, they were reported as densely populated areas in a multidimensional space of community composition, indicating their unfixed demarcation. Analyzing the healthy human gut microbiomes of 495 datasets derived from four continents, Mobeen et al. identified three enterotypes by intercontinental comparisons (*Prevotella*, *Bacteroides*, *Bifidobacterium*), while at the intracontinental level, there are two in America (*Bacteroides*, *Ruminococcaceae*), four in Europe (Faecalibacterium, *Bacteroides*, *Prevotella*, *Clostridiales*) and two in Asia (*Prevotella*, *Bacteroides*/*Bifidobacterium*). These enterotype differences demonstrated the significance of geographical distance on the microbial composition combined with other factors (gender, age, nutrition, etc.) [14]. The process of enterotyping enables the stratification of the human gut microbiome and, therefore, the dimensional reduction of global microbiome variation into a few categories [15].

Figure 1 shows a taxonomic diagram of the most abundant bacteria of the gut microbiome.

### 2.2. Factors Affecting Gut Microbiome Composition

#### 2.2.1. Birth Delivery and Infancy

Microbiome composition is influenced by many factors already in the neonatal period. Intrapartum antimicrobial therapy in caesarean and vaginal delivery has been associated with dysbiosis in infants, while breastfeeding has shown a beneficial influence on the infant microbiome composition [16]. Low-dose penicillin in infants can lead to a disturbance of the microbiota during maturation, inducing long-term metabolic alterations such as adult obesity, nonalcoholic fatty liver disease (NAFLD), and visceral fat accumulation [17]. The microbiota of vaginally delivered infants appears to be more diverse in terms of bacteria species (*B. longum*, *B. catenulatum*) than the microbiota of neonates delivered by caesarean delivery [18], and some of these differences in microbiome diversity have been noticed even beyond infancy [19]. The gut microbiota of breastfed infants is considered more favorable, having a higher richness and diversity of *Bifidobacterium* spp. and a lower number of *C. difficile* and *E. coli* than formula-fed infants [10,20].

#### 2.2.2. Aging

The human gut microbiome changes with aging. Various reasons in elderly people could be attributed to a different gut microbiota compared to healthy adults, such as medication usage, recurrent infections and hospitalizations, weaker immunity systems, and dietary and sleeping habits. This leads to a reduced phylogenetic diversity which is considered to be a part of the progression of numerous metabolic diseases in the elderly [21,22]. Their microbiota composition is susceptible to unvaried nutrition habits, digestion, absorption, and immunity changes [10], and therefore, it is not surprising the observed decrease in anaerobic bacteria (*Bifidobacterium* spp.) and an increase in Clostridium and Proteobacteria [23]. However, these microbiota differences also depend on the geographical location, while it remains undetermined whether microbiota alterations are the cause or repercussion of host aging [21].

#### 2.2.3. Antibiotics

Widespread usage of antibiotic therapy poses a global threat to the health system. Antibiotics alter the gut microbiome inducing the appearance of harmful resistant strains and a cluster of antibiotic resistance genes while decreasing the abundance of protective members and encouraging the overgrowth of opportunistic pathogens in the microbiome [24,25]. The study by Dethlefsen et al. [26] revealed a rapid and significant loss of gut microbiota diversity and a composition change caused by ciprofloxacin administration. They also noticed that the gut microbiota composition never returned to the initial state by the end of the experiment and concluded that antibiotic usage could cause an altered steady microbiome state of unknown consequences. The research conducted by Pérez-Cobas et al. revealed that the impact of antibiotic treatment on the gut microbiome is closely linked to the structure, functions, resistance genes, and characteristics of the microbial community [24]. Treatment with a fluoroquinolone (moxifloxacin) showed a high abundance of the families *Lachnospiraceae* and *Ruminococcaceae*, while in the first days of treatment, there was a registered reduction in the *Bacteroides* genus with a trend of abundance increase in the following days of antibiotic therapy. Treatment with clindamycin presented a high abundance of Enterobacteriaceae and an increase in *Bacteroides*. Treatment with cefazolin/ampicillin/sulbactam resulted in an increase in *Parabacteroides* (Bacteroidetes phylum) and a later increase in facultative anaerobic families, *Enterobacteriaceae* (Proteobacteria) and *Enterococcaceae* (Firmicutes). In the end, treatment with amoxicillin showed that both genera (*Escherichia* genus, *Faecalibacterium* genus) were significantly affected by antibiotic treatment increasing resistant bacterial taxa of the *Bacteroides* genus. Rifaximin treatment in cirrhotic patients has been associated with cognition improvement, reduction in endotoxemia, increase in serum fatty acid metabolites, and bile acid (BA) composition changes with anti-inflammatory promotion [27]. Notable alterations in the microbiome composition and functionality have been noticed in patients with cirrhosis who received chronic rifaximin or norfloxacin treatment to prevent the recurrence of hepatic encephalopathy (HE) [28].

#### 2.2.4. Body Mass Index (BMI), Physical Activity, Dietary Habits, and Sociodemographic Aspects

BMI levels have been represented as reliable predictors for gut microbiota dysbiosis [10]. Elevated levels of Firmicutes (*Ruminococcaceae*) and depleted levels of Bacteroidetes (*Bacteroidaceae*, *Bacteroides*) have been observed in the obese population [29,30]. Conversely, patients suffering from anorexia nervosa exhibited a marked elevation in *Enterobacteriaceae* and *Methanobrevibacter smithii* within their gut microbiota compared to healthy controls. Simultaneously, there was a notable reduction in the abundance of genera such as *Roseburia*, *Ruminococcus*, and *Clostridium* [31].

A study conducted on children aged 7–18 indicated a significant enrichment of the *Proteobacteria* phylum with a higher BMI level and a significant enrichment of the *Firmicutes* phylum (*Clostridiales*, *Roseburia*, *Lachnospiraceae*, *Erysipelotrichaceae*) with frequent exercise [32]. Also, the study by Koliada et al. [33] showed a gradual increment in *Firmicutes* and a reduction in *Bacteroidetes* with an increase in BMI in the Ukrainian adult population. The ratio of *Firmicutes*/*Bacteroidetes* showed an increase with higher BMI and was significantly associated with BMI.

Green tea, caffeine, coffee, adherence to a Mediterranean diet, and the consumption of certain polyunsaturated fatty acids (PUFAs) like omega-3 have been found to promote a beneficial impact on the composition of the gut microbiota. In contrast, the intake of saturated fatty acids, fructose, and advanced glycated end products has been associated with harmful changes to the gut microbiota [22].

Research indicates that the consumption of alcohol leads to a rise in the proportion of *Proteobacteria*, *Enterobacteriaceae*, and *Streptococcus* compared to other gut microbes. Concurrently, there is a decrease in the presence of *Bacteroides*, *Akkermansia*, and *Faecalibacterium* in response to alcohol consumption [34].

Dietary fructose has been associated with dysbiosis by the mechanism of a reduction in phylogenetic diversity, worsening endotoxemia and increasing hepatic Toll-like receptor (TLR) expression [35]. Furthermore, a decline in the consumption of glutamine, tryptophan, and zinc, or an escalation in the intake of fat, alcohol, and food additives, have been linked to heightened IP [36,37,38].

Generally, dysbiosis alters the production of short-chain fatty acids (SCFAs) and choline and BA metabolism. Also, it is associated with a higher abundance of lipopolysaccharide (LPS)-containing bacteria, increased bacterial-derived ethanol, increased IP, and upregulation of inflammatory processes [22]. The polysaccharide fermentation in the large bowel generates SCFAs (acetic, propionic, butyric). Diets high in fiber (plant-based foods, Mediterranean diet) are associated with increased levels of fecal SCFAs [39,40]. SCFAs represent a source of energy (hepatic gluconeogenesis or lipogenesis) and regulate inflammation processes [22,41].

The sociodemographic determinants may significantly shape the gut, as those who live in lower socioeconomic strata, common among end-stage liver disease patients, experience economic constraints that extend to dietary patterns influenced by affordability, safe food handling, and living conditions including water quality, and these aspects hold relevance even in more developed nations [11,14].

#### 2.2.5. Proton Pump Inhibitors

Proton pump inhibitors (PPIs) belong to one of the most used groups of drugs. Their usage has been associated with an increased risk of enteric infections which are based on consequential gut microbiome changes. Imhann et al. [42] demonstrated an increase in the order Actinomycetales, families *Streptococcoceae* and *Micrococcoceae*, genus *Rothia*, and species *Lactobacillus salivarius* in participants using PPIs. Also, in the general population, class Gammaproteobacteria, the family *Enterococcoceae*, and the genera *Streptococcus*, *Veillonella*, and *Enterococcus* were significantly increased. Freedberg et al. [43] found significant changes during PPI use associated with C. difficile infection (increased *Enterococcaceae* and *Streptococcaceae*, decreased *Clostridiales*) and taxa associated with small intestinal bacterial overgrowth (SIBO) (increased *Micrococcaceae* and *Staphylococcaceae*). Furthermore, Jackson et al. [44] demonstrated a higher abundance of *Lactobacillales*, mostly *Streptococcaceae*, in PPI users and concluded that bacterial families increasing with PPI consummation are more likely to originate from the pharynx than the gut. In a study conducted by Bajaj et al. [45], it was found that a 14-day course of 40 mg/day of omeprazole led to noteworthy changes in gut microbiota composition and function in both cirrhosis patients and healthy controls. The study involved 15 patients with cirrhosis, of which 8 had chronic hepatitis C (CHC), 2 had alcohol-related cirrhosis, and 5 had a combination of CHC and alcohol-related cirrhosis. These patients were compared to 15 age-matched healthy individuals who tested negative for *H. pylori* on serology. Following omeprazole use, there was a significant rise in serum gastrin levels compared to the baseline in both cirrhosis patients and controls. However, there was no significant difference in gastrin concentrations between the two groups at baseline or after omeprazole treatment. The results demonstrated a substantial increase in the relative abundance of *Streptococcaceae* after proton pump inhibitor (PPI) therapy compared to the baseline in both cirrhosis patients and controls. Additionally, there was a statistically similar relative change in *Streptococcaceae* after PPI therapy between both groups and a significant positive correlation between serum gastrin and *Streptococcaceae*. Also, in patients with cirrhosis, a reduction in autochthonous bacterial abundance (*Lachnospiraceae*, *Ruminococcaceae*) was observed and no significant change in other major families in the control group. These significant microbiota changes and reduction in gastric acidity could stimulate SIBO or *Clostridioides difficile* infection [45,46]. Treatment with PPI has been described as a risk factor for HE and spontaneous bacterial peritonitis (SBP) in cirrhosis patients with ascites [47], with the risk for HE increasing with the dose of PPI [48].

#### 2.2.6. Nonselective Beta-Blockers (NSBBs)

The gastrointestinal tract’s motility, secretion, and immune function are regulated by the sympathetic nervous system. In advanced stages of cirrhosis, this system becomes more active, especially in the splanchnic area, as a response to the overall activation of vasoconstrictor systems caused by splanchnic vasodilation. Consequently, there is an increase in the levels of catecholamines, which may promote the growth of commensal bacteria [49]. Additionally, high sympathetic tonus defers peristalsis and immunosuppresses by inhibition of chemotaxis and bacterial phagocytosis [50]. NSBBs function by reducing sympathetic activity, which leads to a portal pressure decrease and offers protection against variceal hemorrhage in cirrhosis. They achieve this by blocking β-1 adrenoreceptors, resulting in a cardiac output decrease, and by blocking β-2 adrenoreceptors, leading to an increase in splanchnic vasoconstriction [51]. A meta-analysis by Senzolo et al. [52] evaluated the role of NSBBs in preventing SBP in patients with liver cirrhosis and ascites and revealed a statistically significant difference of 12.1% in favor of propranolol in preventing SBP, independent of the hemodynamic response. Treatment with NSBBs in cirrhotic patients was associated with an increment in intestinal transit and a reduction in intestinal bacterial overgrowth, IP, and bacterial translocation (BT) [46,53,54]. In a study with cirrhotic ascitic rats treated with propranolol, Perez-Paramo et al. showed less BT (15 vs. 58%) and SBP incidence (8 vs. 33%) compared with the control group [54]. The study by Forslund et al. demonstrated an association between beta-blocker use and the intestinal enrichment of the bacterial genus Roseburia [55]. Reiberger et al. [56] included 50 cirrhotic patients with portal hypertension (PH) (72% male, 18% ascites, 60% alcoholic etiology) and evaluated the IP and BT before and after NSBB treatment. The results of their study showed that NSBB treatment led to a mean reduction in the hepatic venous pressure gradient (HVPG) of −19%, with 51% of patients achieving a hemodynamic response, and a significant decrease in IP expressed through a reduction in LPS-binding protein and interleukin (IL)-6 plasma levels, not only seen in hemodynamic responders but also nonresponders as well. Furthermore, NSBBs have been associated with the mitigation of systemic inflammation by reducing mesenteric venous congestion and directly through IP decrease [50,57,58,59]. Also, Mookerjee et al. [58] revealed that NSBB treatment in cirrhosis was associated with lower grades of acute-on-chronic liver failure (ACLF) and with patient improvement associated with a significantly lower white cell count. Considering all the previously mentioned information regarding the effects of NSBBs, an increasing body of evidence suggests that the benefits of NSBBs in cirrhosis patients may not solely arise from the reduction in portal pressure. There might be a direct effect, possibly affecting intestinal transit time or influencing the integrity of the bowel mucosa [57].

#### 2.2.7. Statins

Statins are widely prescribed medications for lowering cholesterol, particularly low-density lipoprotein cholesterol. In a study conducted by Khan et al. [60], 15 untreated hypercholesterolemic patients and 27 hypercholesterolemic patients treated with atorvastatin were included and compared with 19 healthy subjects. They observed an increase in the relative abundance of *Proteobacteria* in untreated hypercholesterolemic patients compared to treated and healthy groups. In the atorvastatin-treated hypercholesterolemic patients, they found a greater abundance of the anti-inflammation-associated bacteria (*Faecalibacterium prausnitzii*, *Akkermansia muciniphila*, and genus *Oscillospira*) and a reduction in the proinflammatory species *Desulfovibrio* compared with the untreated hypercholesterolemic patients. Moreover, the group of patients treated with atorvastatin showed a reduced bacterial diversity, indicating that this treatment might have a selective effect in restoring the relative abundance of several dominant and functionally significant microbial taxa that were disturbed in hypercholesterolemic patients. The review study by Sun et al. [61] has accentuated that statins can modulate the production levels of gut-microbiota-derived metabolites (SCFAs, BAs, trimethylamine (TMA) N-oxide, LPS) by altering various signaling pathways.

#### 2.2.8. Diet, Probiotics, Prebiotics, and Postbiotics

According to a study involving a global cohort of cirrhosis patients, comprising 157 individuals from the U.S. (with 48 controls, 59 compensated, and 50 decompensated) and 139 from Turkey (with 46 controls, 50 compensated, and 43 decompensated), a significant correlation was found. The study revealed that increased microbial diversity in the gut was independently associated with a reduced risk of 90-day hospitalizations. The most important differences between the two international cohorts were that U.S. patients with cirrhosis had more men, greater rifaximin/lactulose usage, higher hepatitis C virus (HCV)/alcohol origin, and generally higher coffee intake, while the Turkish cohort had a higher intake of tea, fermented milk, and chocolate. Consuming a diet abundant in fermented milk, vegetables, cereals, coffee, tea, and chocolate was found to be associated with greater microbial diversity in the gut. Conversely, factors such as a higher Model for End-Stage Liver Disease (MELD) score, lactulose use, and carbonated beverage consumption were linked to lower microbial diversity. Although both cohorts had similar MELD scores, the Turkish cohort had a lower rate of hospitalizations compared to the American cohort, which could be partly attributed to the higher proportion of U.S. patients on lactulose. The Turkish cohort, which entirely had a significantly higher microbial diversity than Americans, showed a lower risk of 90-day hospitalizations [62].

Lactulose presents a nonabsorbable disaccharide with two described mechanisms: a laxative by increasing the volume of stools and a prebiotic by acidifying and modifying the colonic microbiota [46,63]. Lactulose withdrawal in patients with a history of overt HE showed minimal effect on stool composition after 14 days in a way that only *Faecalibaterium* spp. decreased [64].

Probiotics and synbiotics (a combination of probiotics and prebiotics) have been widely used in numerous trials referring to CLD. In a study involving cirrhotic patients with minimal HE, the daily use of Lactobacillus GG (LGG) capsules for eight weeks demonstrated notable results. Compared to the placebo group, the intervention group showed a significant decrease in pathogenic taxa associated with worse cognition (*Enterobacteriaceae* and *Porphyromonadaceae*) and an increase in beneficial autochthonous taxa (*Lachnospiraceae* and Clostridiales XIV) [63,65]. Emerging scientific exploration is focusing on the intriguing realm of postbiotics within the gut microbiome. These are dynamic compounds engendered through the metabolic processes of probiotic microorganisms within the intestinal microbiome. These compounds wield promising potential for bolstering the host’s well-being, encompassing enhancements to immune retorts, dampening inflammation, fortifying digestive processes, and upholding microbiome equilibriums. What sets postbiotics apart is their enhanced safety profile, as they circumvent the necessity of introducing live microorganisms or probiotic cultures. While postbiotic research is still unfolding, preliminary cues hint at their promise as a plausible substitute or complement to probiotics and prebiotics for nurturing a resilient gut microbiome and holistic vitality. Several illustrations of postbiotics comprise fragments of microbial cells, SCFAs, extracellular polysaccharides, cellular lysates, vitamins, and teichoic acid [6,10].

Figure 2 shows the shares of the kingdoms that make up the microbiome and the factors that influence it.

## 3. The Gut Microbiome in Chronic Liver Disease

### 3.1. Gut Microbiome Changes in Chronic Liver Disease

In a comparative study of the mucosa-associated colonic microbiome, researchers examined alcoholics with or without alcoholic liver disease (ALD) and healthy controls. The findings revealed a state of gut microbiome dysbiosis in a subset of alcoholics, characterized by reduced levels of *Bacteroidetes* and elevated levels of *Proteobacteria* compared to healthy individuals [66]. ALD-related cirrhosis exhibited an altered fecal microbiome with reduced *Bacteroidaceae* and a notable increase in *Prevotellaceae* compared to healthy individuals. Interestingly, bacterial deoxyribonucleic acid (DNA) from *Enterobacteriaceae* was found to be the most prevalent in the cirrhotic liver, contrasting with healthy volunteers. These findings indicate significant microbiota changes in the gut and liver associated with ALD-related cirrhosis [67,68]. In a human study of cirrhotic patients with hepatocellular carcinoma (HCC), the fecal microbiota dysbiosis is characterized by an overgrowth of *E. coli* [69]. *Helicobacter* species 16S rDNA was detected in 8 of 20 liver samples of HCC, whereas no evidence of *Helicobacter* could be found in patients without malignancy [70]. A comparison of the DNA sequences suggested a great similarity with *H. pylori* species, while the presence of Helicobacter species in HCC tissue speaks in favor of a possible carcinogenic effect [70]. The gut microbiome can form an immunosuppressive environment in the liver by controlling hepatocytes through Gram-negative bacteria/LPS, which interact with TLRs, especially TLR4, on Kupffer cells (KCs) and hepatic stellate cells (HSCs). Activated KCs and HSCs initiate a proinflammatory and profibrotic process which is further mediated by cytokines (IL-1, IL-6, tumor necrosis factor (TNF-α)), leading to the accumulation of polymorphonuclear myeloid-derived suppressor cells. Also, the cluster of differentiation (CD) 14 expression percentage in the cirrhotic liver is significantly higher, which suggests that bacteria may have a role as inducers of the CD14-mediated proinflammatory process, finally leading to cirrhosis and, over time, creating a favorable condition for potential tumor growth [22,68,71].

#### 3.1.1. How Chronic Liver Disease Affects the Composition of the Gut Microbiome

The portal vein, biliary ducts, and enterohepatic recirculation represent the pathways through which the liver communicates with the gut. The portal vein transfers nutrients and metabolic products from the gut microbiome, including microbe-associated molecular patterns (MAMPs) to the liver, and secondary BAs from the gut enter the enterohepatic recirculation ending up again in the liver. The liver excretes primary BAs, immunoglobulin A (IgA), and some antibacterial substances which are delivered into the gut via the biliary ducts. In addition to this, liver-derived metabolites (such as very-low-density lipoprotein (VLDL)) or some inflammatory mediators reach the bowels via the systemic circulation [72]. There are two main producing pathways of BAs—“neutral” regulated by CYP7A1 and “acidic” regulated by CYP27A1—and both produce primary BAs cholic (CA) and chenodeoxycholic (CDCA) in the liver [63]. Conjugated primary BAs (CA, CDCA) undergo various microbial modifications such as dehydroxylation by colonic 7α-dehydroxylating bacteria (*Ruminococcaceae*, *Lachnospiraceae*, and *Blautia)* becoming secondary BAs (deoxycholic (DCA), lithocholic (LCA)) [73,74]. Bile acids (BAs) play a vital role in lipid absorption and have a significant shaping impact on intestinal microbiomes. Their antimicrobial effects are achieved through the farnesoid-X receptor (FXR) activation, leading to the production of antimicrobial peptides that help in selecting and maintaining a healthy gut microbial community [75,76,77]. By binding to FXR in the enterocytes, BAs impact different metabolic and inflammatory processes such as the inhibition of bacterial overgrowth and deactivation of endotoxins [78].

Inflammatory mediators released along the development and worsening of CLD suppress the synthesis of primary BAs through the CYP7A1 pathway, causing a reduced concentration of BAs in the intestines which creates a susceptible milieu for pathogenic and proinflammatory microbiome members such as *Porphyromonadaceae* and *Enterobacteriaceae* [79]. Consequently, the metabolism switches to an alternative pathway that uses sterol-27-hydroxylase (CYP27A1) to synthesize mostly CDCA but not CA [80]. The decreased delivery of CA to the colon results in the decreased production of secondary deoxycholic acid (DCA, by 17-α-dehydroxylation, mostly from the *Clostridium* genus) which exerts the highest antimicrobial activity among all BAs [81,82,83,84]. As a result, there are fewer primary BAs for conversion to secondary BAs by the *Clostridium* genus which is potentially reduced and paves the way to the overgrowth of pathogenic families such as *Enterobacteriaceae* leading to dysbiosis [63]. Knowing that BAs act preventively in bacterial overgrowth and promotionally in maintaining epithelial cell integrity, decreased BA intraluminal concentration may promote dysbiosis and bacterial overgrowth [46]. The study by Kakiyama et al. [85] reported a decrease in total fecal BA concentration and a reduction in the ratio of secondary to primary BAs along the worsening clinical stages of liver cirrhosis. Moreover, the study revealed a reduction in naturally occurring genera and an overgrowth of *Enterobacteriaceae* within the microbiome of cirrhotic patients, accompanied by a notable increase in serum bile acids (BAs) compared to control subjects. Additionally, the naturally occurring genera exhibited a positive correlation with secondary BAs and the ratios of secondary to primary fecal BAs, while potentially pathogenic genera demonstrated a correlation with primary BAs [85].

#### 3.1.2. How the Altered Composition of the Gut Microbiome Affects the Liver

Dysbiosis affects normal liver physiology by upregulating hepatic lipogenesis and triglyceride storage, and, in contrast, reducing lipid oxidation leading to hepatic steatosis. The activation of TLR4 and the reactive oxygen species (ROS) induces hepatic inflammation and fibrosis [22]. Dysbiosis also leads to damage of the mucosal barrier, resulting in increased IP, a pathophysiological development with profound consequences on liver health.

Almost 40 years ago, Bjarnason et al. discovered higher IP in nonintoxicated alcoholic patients than controls by a chromium-51 absorption test [86]. The study by Keshavarzian et al. [87] showed increased IP in alcoholics with chronic liver disease compared to alcoholics with no liver disease and nonalcoholics with liver disease by measuring the urinary excretion of lactulose and mannitol after oral administration. They concluded that a “leaky” gut may be a necessary cofactor for the development of CLD in chronic alcoholics. Chen and Schnabl also determined increased IP in ALD patients [67]. Several research studies [88,89,90] conducted on rats have consistently shown that acute alcohol consumption leads to elevated IP, endotoxemia, and liver damage. It has been established that alcohol-induced gut hyperpermeability and endotoxemia occur before the development of steatohepatitis, serving as a critical trigger for alcoholic steatohepatitis. The study by Miele et al. [91] observed that patients with NAFLD exhibited significantly higher gut permeability compared to healthy individuals. Additionally, in patients with NAFLD, both gut permeability and the prevalence of SIBO correlated with the severity of steatosis, although not with steatohepatitis. Verdam et al. [92] found significantly elevated plasma immunoglobulin G (IgG) levels against endotoxin in patients with biopsy-proven nonalcoholic steatohepatitis (NASH) compared to individuals with healthy livers. Also, these IgG levels have been found to progressively increase with the NASH grade, suggesting an association between long-term endotoxin exposure and NASH severity. A “leaky” gut phenotype in ALD was also represented by animal models of ethanol administration [93]. Acetaldehyde primarily disrupts the integrity of adherens and tight junctions through a mechanism that relies on phosphorylation. However, there are several gastrointestinal mucosal protective factors, such as epidermal growth factor, glutamine, zinc, oat bran, and probiotics, which counteract the adverse effects of ethanol and acetaldehyde on IP. These protective factors play a crucial role in preventing the occurrence of endotoxemia and liver damage induced by ethanol/acetaldehyde [93].

Endotoxin LPS represents a cell component of Gram-negative bacteria which has been known to induce inflammation, metabolic syndrome, nonalcoholic hepatic steatosis, and fibrosis in the liver [94,95,96]. Microbial fragments such as LPS, lipopeptides, bacterial DNA, and peptidoglycan represent pathogen-associated molecular patterns (PAMPs) which transit through the portal vein into the liver and modulate numerous functions by metabolite-dependent pathways mediated by TLRs [97,98]. LPS interacts particularly with TLR4 on KCs and HSCs to trigger proinflammatory and profibrotic pathways resulting in the production of inflammatory cytokines, including IL-1, IL-6, and TNF-α, which affect pathogenesis, progression, and the development of the immune response in the liver [99,100]. The presence of LPS and other gut-microbiome-derived TLR ligands has been linked to adipose tissue inflammation, leading to alterations in the secretion of various adipokines (such as adiponectin, IL-6, leptin, and resistin) that further contribute to liver inflammation [22,101]. This inflammatory response in adipose tissue, accompanied by tissue expansion, dysfunction, and inflammation, plays a significant role in NAFLD development. Furthermore, LPS has been shown to promote the accumulation of lipids in the liver and cause hepatocyte Inflammation [22,102]. Notably, individuals with NAFLD and nonalcoholic steatohepatitis (NASH) have been found to exhibit higher levels of LPS in both the peripheral circulation and liver compared to healthy controls [103].

Figure 3 represents the communication pathways of the gut–liver axis and the cascade of consequent “leaky” gut events.

As for ALD, studies showed that LPS and ethanol have a combined effect on the induction of liver injury. Beginning with translocation from the intestinal lumen, LPS arrives via the portal circulation in the liver and causes activation of KCs through TLR4 or CD14 signal pathways [67]. Additionally, LPS induces proinflammatory cytokine production and a reduction in three components: adrenergic stimulation, ROS production, and IL-10-mediated protection [93,104,105,106]. Studies have shown that the influence of LPS includes HSCs, KCs, liver sinusoidal endothelial cells (LSECs), hepatocytes, and neutrophils. LPS causes the stimulation of cytokine and chemokine release in LSECs and an ethanol-induced collagen secretion increment in HSCs. LPS-binding protein presents LPS to CD14 and then CD14 binds specifically to LPS, enabling interaction with TLR4 [93,107,108,109,110]. Furthermore, an elevation of bacterial DNA has been found in the plasma of patients with alcohol-related cirrhosis, which has the potential to contribute to ALD by TLR9 recognition and LPS induction of liver injury [67,111,112].

Inflammasomes are cytosolic multiprotein oligomers which represent a part of the innate immune system. TLR signaling in the mucosa promotes the production of inflammasomes, causing further proinflammatory and profibrotic reactions by other mediators (caspase-1, IL-1β, IL-18) [113]. While some studies found significantly higher levels of the inflammasome nucleotide-binding oligomerization domain-like receptor (NLR) family pyrin domain containing 3 (NLRP3) in NASH patients compared to simple steatosis [114], others have shown an association of more aggressive liver disease with inflammasome absence. The Western diet (a combination of a high-fat and high-carbohydrate diet) and the lack of the NLRP3-inflammasome have been associated with an increment in liver injury, an abundance of Proteobacteria and Verrucomicrobia, and higher BT and TLR activation [115]. Also, NLRP3 has been presented as a potential target for the manipulation of the gut microbiota that may interfere with the progression of liver injury in NAFLD [115].

Choline-deficient diets have been associated with hepatic steatosis [116]. The role of the gut microbiota has been implicated in the imbalance of choline metabolism after a shown association of NAFLD with lower levels of choline and higher levels of TMA in the blood [117]. Considering that about 10–15% of bacterial species need choline to synthesize phosphatidylcholine as the component of their membrane, intestinal dysbiosis and bacterial overgrowth cause increased requirements for choline and thus potential choline deficiency [118,119].

Numerous changes in the gut microbiome in chronic liver disease arise from various factors (alcohol consumption, drugs, malnutrition, genetics, viral infections, autoimmune disorders, etc.). Furthermore, intestinal dysbiosis could promote the dysfunction of tight junction proteins between intestinal epithelial cells by inducing intestinal inflammation, consequently causing increased IP or a “leaky” gut. IP allows BT, microbial products, and endotoxins (LPS) to cause inflammatory processes in the liver tissue and therefore liver disease progression [67,120,121]. There are several diagnostic tools for detecting BT. Direct measures of IP are dual sugar probes (e.g., lactulose/mannitol) as a gold-standard method which includes the usage of two sugar controls for nonmucosal factors, Cr-EDTA (^51^Cr-labelled ethylenediaminetetraacetic acid) and PEG (polyethylene glycol), which assess the whole intestine, FITC-dextran, and transcutaneous fluorescence. These tests are time-consuming and require overnight fasting, ingesting the sugar probe(s), and drinking large amounts of water in a short period, which is quite demanding for patients with severe liver disease [122]. Therefore, alternative methods include systemic markers of BT as an indirect assessment of IP. LPS measurement could be considered as a surrogate marker of BT, but its value is influenced by various variables (physiological, immunogenetic, microbiological) and has a short half-life. Another method of detecting BT is a measurement of LPS-binding protein which is produced by the liver in response to bacteriaemia and has a longer half-life, but its value only determines the translocation of Gram-negative bacilli and is increased in infective episodes as an acute phase protein. Polymerase-chain-reaction-based detection of bacterial DNA detects Gram-positive cocci and Gram-negative bacilli, has a longer half-life, and also predicts clinical outcomes but has a variable detection and poor validation procedure. Zonulin, a protein synthesized in intestinal and liver cells, which is involved in the disassembly of tight junction proteins as a regulator of IP, has shown a correlation between increased IP as measured by a dual sugar probe, but its diagnostic validity has been questioned recently. An increase in intestinal fatty-acid binding protein (FABP) in the systemic circulation has been correlated with increased IP. The intestinal FABP method is a readily accessible assay conducted on serum samples. However, its findings are more closely associated with epithelial damage rather than indicating IP increment [122,123].

### 3.2. The Gut Microbiome in Different Etiologies of Chronic Liver Disease

Numerous studies have investigated the gut microbiome composition in different groups of individuals, ranging from healthy subjects to those with NAFLD at various stages. Despite variations in study design, methodologies, and clinical criteria, these investigations consistently reveal distinguishable differences in the gut microbiome between healthy controls and individuals with hepatic steatosis and NASH. However, as Pezzino et al. [124] pointed out, the gut microbiome may vary between demographic groups and stages of NAFLD. Also, different molecular approaches used for bacterial classification to the species level and the defining methodology of NAFLD stages in various studies contribute to these variations. Therefore, there are various studies with some opposite results in the relative abundances of *Bacteroidetes*, *Firmicutes*, and *Ruminococcus* between healthy controls and patients with NAFLD.

The presence of *Proteobacteria*, particularly *Klebsiella pneumonia* and *Escherichia coli*, has been linked to the fermentation of ethanol from dietary carbohydrates, leading to the production of fatty acids and oxidative stress in the liver. These factors are considered significant contributors to the development of NAFLD and NASH [125,126]. Choline, as a precursor of phosphatidylcholine, is necessary for VLDL synthesis and excretion, while the lack of it results in a reduction in VLDL release and an increase in liver triglyceride levels [127]. Furthermore, around 10–15% of bacterial species consume choline for phosphatidylcholine production as a component of their membrane, whereas bacterial overgrowth can lead to choline deficiency. Also, the gut microbiome is well known for the conversion of choline to TMA, which can be oxidized by hepatic monooxygenases, leading to the production of trimethylamine N-oxide. Its elevated levels in the liver cause hepatic inflammation and adverse effects on glucose metabolism by increasing insulin resistance and decreasing glucose tolerance, which all together potentiates the development of NAFLD [77]. Acetate, propionate, and butyrate make up more than 90% of the SCFAs in the digestive tract, and they are produced by the gut microbiota from indigestible starch and fiber in the diet. SCFAs contribute to the onset of NAFLD by inducing enteroendocrine mucosal cells on the release of the gut hormone peptide YY, which slows intestinal transit time and increases nutrient absorption resulting in lipid liver accumulation. Propionate and butyrate act in the process of hepatic autophagy which enables the hydrolysis of triglycerides and the release of free fatty acids for mitochondrial β-oxidation [128,129].

An anaerobic bacterium *Akkermansia muciniphila* (type *Verrucomicrobia*), found in the gastrointestinal tract in about 80% of people, produces acetates and propionates and therefore provides energy for intestinal cells. Studies indicate a favorable effect of *A. muciniphila* on the intestinal barrier by showing how an increase in the *A. muciniphila* amount in mice has been associated with intestinal barrier improvement, leading to a reduction in proinflammatory LPS and better glucose control [130,131].

Shen et al. [132] analyzed the gut microbiome composition in a group of 47 adults (25 with NAFLD and 22 healthy controls) and found a lower diversity and concentration of *Prevotella* and a higher concentration of *Proteobacteria* and *Fusobacteria* in individuals with NAFLD. Their study indicated that the increased level of the genus *Blautia*, the family *Lachnospiraceae*, the genus *Escherichia*/*Shigella*, and the family *Enterobacteriaceae* may be a primary contributor to NAFLD progression. Wang et al. [133] included a group of 126 nonobese adults (43 with NAFLD on ultrasound and 83 healthy controls) and found a lower diversity, lower concentration of *Firmicutes* and a higher concentration of *Bacteroidetes* and Gram-negative species in individuals with NAFLD. In a group of 75 adults (25 with biopsy-proven nonalcoholic steatosis, 25 with biopsy-proven NASH, and 25 healthy controls), Tsai et al. [134] showed at the phylum level that NAFLD and NASH patients had higher levels of *Bacteroidetes* and lower levels of *Firmicutes* than healthy individuals, which corresponds to the Wang et al. [133] and Wong et al. [135] studies, as they both examined Asian populations. Unlike the previously mentioned studies [133,134,135], a study by Mouzaki et al. [136] showed a connection between the percentage of *Bacteroidetes* in the stool and the presence of NASH, being independent of diet and BMI.

The composition of the gut microbiome in NAFLD appears to differ depending on the stage of liver fibrosis. In a study by Loomba et al. [137], they examined the gut microbiome composition in 86 adults with biopsy-proven NAFLD, 72 of whom had mild hepatic fibrosis (stage 1–2), and 14 had advanced hepatic fibrosis (stage 3–4). Their findings revealed that in mild/moderate NAFLD, the most abundant organisms at the species level were *E. rectale* (2.5% median relative abundance) and *B. vulgatus* (1.7%). On the other hand, in cases with advanced fibrosis, the most abundant organisms were *B. vulgatus* (2.2%) and *E. coli* (1%). Additionally, the study observed a decrease in Gram-positive *Firmicutes* and an increase in Gram-negative *Proteobacteria* (including *E. coli*) in patients with advanced NASH fibrosis. This suggests that the gut microbiota shifts toward more Gram-negative microbes in advanced fibrosis, while *Bacteroidetes* showed a statistically insignificant increase (23.62% in the mild hepatic fibrosis group vs. 28.46% in the advanced hepatic fibrosis group). Loomba et al. [137] suggested that *E. coli* dominance occurs in advanced fibrosis before the appearance of ascites or any signs of liver decompensation and therefore supported the hypothesis that dysbiosis may precede the development of PH. Boursier et al. [138] enrolled 57 patients with NAFLD proven by biopsy (30 patients with F0/1 and 27 with F ≥ 2 fibrosis stage). Their results showed a significant increase in *Bacteroides* and a decrease in *Prevotella* in NASH and F ≥ 2 patients, whereas a significant increase in *Ruminococcus* abundance in F ≥ 2 patients was observed. After conducting a multivariate analysis, they identified three subgroups based on increasing NAFLD severity: low NASH/low fibrosis, high NASH/low fibrosis, and high NASH/high fibrosis. Interestingly, the abundance of *Bacteroides* was independently associated with NASH, while *Ruminococcus* was associated with F ≥ 2 fibrosis.

In their study, Puri et al. [139] investigated alterations in the circulating microbiome of individuals diagnosed with alcoholic hepatitis (AH) with different severity levels. They employed bacterial DNA sequencing to analyze the samples from subjects with moderate AH (*n* = 18) and severe AH (*n* = 19), comparing them to heavy drinking controls (*n* = 19) and nonalcohol-consuming controls (*n* = 20). AH was defined by a combination of hyperbilirubinemia, elevated aspartate aminotransferase levels, and a history of heavy alcohol consumption for at least six months, including the last consumption within six weeks of presentation and without an alternate cause of hepatitis. The severity classification of AH was based on the MELD score. Patients with MELD scores exceeding 20 were categorized as having severe AH, while those with scores lower than 20 as moderate AH. Heavy drinking controls included subjects without clinical findings suggestive of AH and who had normal bilirubin and liver enzymes, whereas nonalcohol-consuming controls had no clinical, laboratory, or imaging evidence of liver disease. The results showed a significant decrease in *Bacteroidetes* in the heavy drinking controls, subjects with moderate and severe AH compared to nonalcohol-consuming controls. On the contrary, there was a higher abundance of *Fusobacteria* in all alcohol-consuming groups. Their results also indicated significantly higher endotoxemia in subjects with severe AH. In their study, Yang et al. [140] demonstrated that alcohol-dependent patients displayed reduced intestinal fungal diversity and *Candida* overgrowth, whereas *Candida dubliniensis* tends to increase in patients with AH and is the most abundant *Candida* species in patients with end-stage alcohol-related liver disease. The process of intestinal fungi overgrowth combined with a dysfunctional gut barrier results in increased systemic levels of β-glucan, inducing a chronic inflammatory liver response.

Chronic hepatitis B (CHB) has been associated with a reduction in butyrate-producing bacteria, while it is enriched in LPS-producing genera [141]. Wang et al. [142] investigated the gut microbial stool composition in CHB patients with low CTP scores (not above 9) compared to healthy controls. They observed a significant increase in five operational taxonomic units belonging to *Actinomyces*, *Clostridium sensu stricto*, unclassified *Lachnospiraceae*, and *Megamonas* in CHB patients, while a significant decrease in units belonging to *Alistipes, Asaccharobacter*, *Bacteroides*, *Butyricimonas*, *Clostridium IV*, *Escherichia/Shigella, Parabacteroides*, *Ruminococcus*, and various other unclassified families. Also, four units (one each belonging to Veillonella and Haemophilus and two to Streptococcus), which were significantly higher in CHB patients with higher CTP scores, showed high correlations with aromatic amino acids phenylalanine and tyrosine. These higher levels of aromatic amino acids indicate impaired phenylalanine and tryptophan metabolism in CHB patients, and overall microbiome changes suggest a potential contribution to CHB progression. Liu et al. [143] demonstrated that patients with hepatitis B virus (HBV)-related HCC have a higher abundance of potential anti-inflammatory bacteria (*Faecalibacterium*, *Pseudobutyrivibrio*, *Lachnoclostridium*, *Ruminoclostridium*, *Prevotella*, *Alloprevotella*, and *Phascolarctobacterium*) and a reduction in proinflammatory bacteria (*Escherichia*/*Shigella*, *Enterococcus*) compared with non-HBV-, non-HCV-related HCC patients. *Lachnospiraceae* showed a beneficial effect on CHB by reducing LPS and BT [144]. According to Lu et al. [145], cirrhotic patients with HBV infection exhibit significant fluctuations in the quantities of various gut microbiota, including *Faecalibacterium prausnitzii*, *Enterococcus faecalis*, *Enterobacteriaceae*, *Bifidobacteria*, and lactic acid bacteria (specifically *Lactobacillus*, *Leuconostoc*, *Pediococcus*, and *Weissella*). Notably, the *Enterococcus* and *Enterobacteriaceae* levels are elevated compared to healthy individuals.

In CHC patients, *Enterobacteriaceae* and *Bacteroidetes* are mostly found to increase, while *Firmicutes* decreased. CHC infection induces LPS elevation, suggesting BT and inflammation during disease progression, whereas antiviral treatment (ribavirin and immune modulator pegylated interferon) increases the production of BAs which has a beneficial effect on the gut microbiota [141]. Sultan et al. characterized the gut microbiota structure in newly diagnosed HCV-infected patients before any antiviral treatment as compared to healthy controls. The analysis revealed an increased prevalence of *Catenibacterium*, *Prevotella*, *Ruminococcaceae*, and *Succinivibrio*, and in the gut of HCV-infected patients. Conversely, *Bacteroides*, *Dialister*, *Bilophila*, *Streptococcus*, *Parabacteroides*, *Enterobacteriaceae*, *Erysipelotrichaceae*, *Rikenellaceae*, and *Alistipes* were present in a lower abundance in these patients’ gut microbiotas [146].

Recent research has linked autoimmune liver diseases, namely autoimmune hepatitis (AIH), primary biliary cholangitis (PBC), and primary sclerosing cholangitis (PSC), with alterations in the commensal microbiota’s composition and abnormal immune system activation triggered by microbial signals, primarily through the gut–liver axis [147]. AIH patients showed a reduction in beneficial anaerobic species such as *Faecalibacterium prausnitzii*, while an increase in the genus *Veillonella* [148,149]. Liwinski et al. also observed a relative increase in the facultative anaerobic genera *Streptococcus* and *Lactobacillus* and an association between the marked depletion of the genus *Bifidobacterium* and a lack of liver inflammation remission [149]. Lou et al. also detected an increased relative abundance of *Veillonella* in AIH patients compared to healthy controls, while, contrary to the previously mentioned studies, they noticed an increased relative abundance of *Faecalibacterium* [150]. They also demonstrated five microbial biomarkers (*Lachospiraceae*, *Veillonella*, *Bacteroides*, *Roseburia*, and *Ruminococcaceae*) for distinguishing AIH patients from healthy controls [150]. In a study by Lv et al. [151], PBC patients showed a depletion of some potentially beneficial bacteria (*Acidobacteria*, *Lachobacterium* sp., *Bacteroides eggerthii*, *Ruminococcus bromii*) and an enrichment in some opportunistic bacterial pathogens (*γ-Proteobacteria*, *Enterobacteriaceae*, *Neisseriaceae*, *Spirochaetaceae*, *Veillonella*, *Streptococcus*, *Klebsiella*, *Actinobacillus pleuropneumoniae*, *Anaeroglobus geminatus*, *Enterobacter asburiae*, *Haemophilus parainfluenzae*, *Megasphaera micronuciformis*, and *Paraprevotella clara*). The loss of *Clostridiales* species was also noticed in PBC patients and a decrease in *Faecalibacterium* in nonresponders to ursodeoxycholic acid, which might be a predictor of the disease prognosis [152]. Several studies [153,154,155,156,157] revealed an increase in the abundance of the genera *Veillonella*, *Enterococcus*, *Streptococcus*, and *Lactobacillus* in patients with PSC, whereas there was a depletion of SCFA-producing anaerobes *Faecalibacterium* and *Coprococcus*.

Based on the results presented in this section, there seems to exist a tendency to increase *Proteobacteria*, *Fusobacteria*, and *Bacteroidetes* and reduce the abundance of *Prevotella* and *Firmicutes* in NAFLD. Pezzino et al. [124] represented a vicious circle of dysfunctions where gut microbiome dysbiosis plays a main role in the disruption of the gut–liver axis, creating a milieu favorable for a progressive form of NAFLD. The mechanisms of pathogenesis include gut barrier impairment and an IP increment resulting in endotoxemia and inflammation and changes in BA profiles and metabolite levels (increasing of endogenous ethanol, reduction in choline levels, dysregulation of SCFA metabolism). Furthermore, studies which included subjects with AH and alcohol-consuming controls mainly showed a significant decrease in *Bacteroidetes* and an increase in *Fusobacteria* in those groups with a propensity for *Candida* overgrowth. The gut microbiome in HBV and HCV infection mainly showed a higher abundance of *Enterobacteriaceae* and a lower abundance of *Bacteroidetes*. Finally, the common characteristics of the gut microbiome in AIH, PBC, and PSC patients include an increase in the genera *Veillonella* and *Streptococcus* and a depletion in the genus *Faecalibacterium*.

### 3.3. The Gut Microbiome in Different Stages of Chronic Liver Disease

Metagenomic technology has been used in identifying the diversity of the human gut microbiome, revealing new genes, and evaluating microbial pathways that can detect functional dysbiosis and the course of some disease states [28].

Qin et al. [158] conducted a study that showcased the promising diagnostic value of microbial markers in liver cirrhosis. They employed quantitative metagenomics and a panel of 15 biomarkers for effectively distinguishing between patients with liver cirrhosis and healthy subjects. Some studies suggested that microbial features are disease-specific. Notably, microbial genes exhibiting high specificity to liver cirrhosis were distinctive from the markers identified in type 2 diabetes [159].

Loomba et al. [137] conducted a study revealing noticeable variations in the gut microbiome composition between individuals with mild fibrosis (stage 1 or 2) and advanced fibrosis (stage 3 or 4). The researchers proposed the usage of a fecal-microbiome-derived metagenomic signature as an additional noninvasive tool for determining the stage of NAFLD alongside current invasive methods. Throughout the progression from mild NAFLD to advanced fibrosis, the phylum-level analysis indicated an increase in *Proteobacteria* and a decrease in *Firmicutes*. Moreover, at the species level, *E. rectale* was found to be the most abundant microorganism in mild fibrosis, while *B. vulgatus* dominated in advanced fibrosis. Furthermore, by identifying 37 microbial species which feature different stages of the disease, they suggested the potential use of microbial markers as a tool in diagnosing and determining the stages of liver disease [137,160]. A study by Rau et al. [161] explored the link between gut microbial changes in NAFLD patients and fecal SCFA concentrations. They indicated that NASH patients had a higher abundance of *Fusobacteria* and *Fusobacteriaceae* compared to NAFLD patients and healthy controls. Also, they found that NAFLD patients had higher acetate ad propionate levels which were associated with lower resting regulatory T-cells (rTregs) and a higher Th17/rTreg ratio in peripheral blood. A higher abundance of SCFA-producing bacteria in the feces of NAFLD patients implies their potential involvement in disease progression. These bacteria can perpetuate low-grade inflammatory responses that affect various peripheral organs, including the liver and circulating immune cells.

Using magnetic resonance imaging, researchers examined the impact of gut bacteria on the gut–liver–brain axis. The study discovered a positive relationship between *Enterobacteriaceae* and *Streptococcacae* and astrocytic changes. Additionally, they observed a connection between *Porphyromonadaceae* and alterations in neuronal integrity and oedema [162].

In their study, Bajaj et al. [163] determined a cirrhosis dysbiosis ratio (CDR) based on the ratio of autochthonous to pathogenic taxa. Alcoholic cirrhotic patients exhibited a unique dysbiosis pattern characterized by a reduced CDR, elevated levels of *Enterobacteriaceae*, and increased endotoxemia when compared to nonalcoholic patients, even though their MELD scores and abstinence status were similar. In patients studied before/after HE development, dysbiosis occurred post-HE (CDR: 1.2 to 0.42, *p* = 0.03). Additionally, in a longitudinal analysis, decreased CDR was found in patients after the occurrence of HE, comparing patients before and after HE development.

#### 3.3.1. Compensated Advanced Chronic Liver Disease (cACLD)

The study by Chen et al. [144] demonstrated a marked decrease in the relative abundance of *Bacteroidetes* (*p* = 0.008) and a high enrichment of *Proteobacteria* (most of them belonging to class *Gammaproteobacteria*) and *Fusobacteria* (*p* = 0.001 and 0.002, respectively) in patients with cirrhosis (a total of 36 patients of which 24 were hepatitis B virus-related and 12 alcohol-related) compared to controls. In the cirrhosis group, the class *Bacilli* (*Streptococcaceae*), affiliated with the phylum *Firmicutes*, *Enterobacteriaceae* and *Pasteurellaceae* within the class *Gammaproteobacteria* were found to be significantly more abundant. Furthermore, a positive correlation trend was observed between *Streptococcaceae* and the clinical stage of liver cirrhosis, as indicated by the CTP score (R = 0.386, *p* = 0.02). Also, the *Fusobacteriaceae* family, as the main component of the *Fusobacteria* class, was predominant in the cirrhosis group (2.7% versus 0.2%). Chen et al. [144] observed, at the family level, a significant increase in *Prevotellaceae* in alcohol-related cirrhosis, while no statistical difference at the phylum and class level was observed between HBV-related and alcohol-related cirrhosis. The family of anaerobic bacteria *Lachnospiraceae*, consisting of genera *Coprococcus*, *Pseudobutyrivibrio*, and *Roseburia*, was found significantly decreased (*p* = 0.004) in the liver cirrhosis group and correlated negatively with the CTP score (R = −0.49, *p* = 0.002). *Lachnospiraceae* participates in the fermentation of carbohydrates into SCFAs and gases (CO_2_ and H_2_O), whereas SCFAs represent nutrients for the colonic epithelium and modulators of colonic pH [144,164,165]. Thus, the reduction in *Lachospiraceae* results in an SCFA decrease, leading toward increased colonic pH and raised intestinal ammonia production and absorption, with hyperammonemia being a crucial pathogenetic factor in HE [166].

#### 3.3.2. Portal Hypertension

PH represents one of the most common repercussions of liver cirrhosis leading to numerous potential complications (ascites, variceal bleeding, HE). It is defined as an increase in the HVPG of > 5 mmHg. Clinically significant portal hypertension (CSPH) develops in the case of HVPG > 10 mmHg. PH-associated splanchnic hyperemia induces an increase in gastrointestinal permeability, BT, and endotoxin levels [167]. On the other hand, it reduces the role of the local immune system in preventing the translocation of bacteria and their products into the systemic circulation and ascites, thus increasing the risk of spontaneous bacteremia and SBP [123]. Inflammatory cytokine levels have been found to differ in the systemic, portal, and hepatic circulations [168,169]. The exacerbated inflow of bacterial products and particles upregulates portal pressure. The development of PH leads to intestinal oedema and the disruption of epithelial integrity, facilitating further BT which induces a proinflammatory response by activating TLRs and NLRs [46]. Furthermore, BT deteriorates the systemic circulation by intensifying peripheral vasodilatation, which has been related to higher levels of TNF-α and worsening PH. A lacking availability of nitric oxide (NO) in the hepatic microcirculation represents an essential factor contributing to an increment in hepatic vascular resistance. Furthermore, LSECs, as part of the reticuloendothelial system, represent one of the most important defense mechanisms against bacteremia and other infections of hematogenous origin. In cirrhosis, the immune function of LSECs is weakened and, in addition to reduced activity in removing viable bacteria, their role in removing bacterial products such as endotoxins or bacterial DNA is also reduced. Thus, the entire BT contributes to the chronic inflammatory response and hemodynamic changes present in cirrhosis [123]. Endothelin 1 (ET-1) plays an important role in the regulation of intrahepatic PH in cirrhosis by stimulating HSC contraction in the liver, where the largest number of ET-1 receptors are located. Released LPS and cytokines during BT stimulate the production of ET-1, which increases portal venous resistance in combination with produced cyclooxygenases during endotoxemia. This mechanism supports an imbalance between the expression of a vasodilator (NO, carbon monoxide) and vasopressor substances (ET-1), leading to a predominance of vasopressors and consequently to an increase in hepatic vascular tone [123,170,171,172,173]. Consequently, the cirrhotic liver has no possibility of vasodilatation in response to a volume flow load (e.g., postprandial), resulting in a sudden increase in portal pressure and thus precipitating intrahepatic endothelial dysfunction that may ultimately lead to variceal bleeding [174].

There are not many studies that have investigated the composition of the gut microbiome in patients with PH, and those that follow below are mostly recent. By investigating different bacterial and inflammatory markers (LPS, FABP-2, IL-6, IL-8) in different blood compartments, Gedgaudas et al. [167] revealed *Proteobacteria* (44%) as being the most dominant phyla in the peripheral circulation, followed by *Bacteroidetes* (27.7%), *Actinobacteria* (18.44%), and *Firmicutes* (9.9%) in patients with PH. Only a relative abundance of *Firmicutes* showed a significant increment in patients with PH compared to healthy individuals. The study found no significant differences in abundant taxa between the hepatic vein and peripheral vein blood compartments in patients with PH. Despite finding an association between the abundance of *Bacteroides*, *Escherichia*/*Shigella*, and *Prevotella* genera with severe PH in both blood compartments, the circulating microbiome profiles were unable to predict the severity of PH (CSPH or severe PH). Moreover, patients with PH exhibited higher levels of LPS, IL-6, and IL-8 compared to healthy controls, with IL-6 and IL-8 levels in the peripheral blood showing correlations with MELD and CTP scores. Another recent study, which included only 32 patients (of which 21 were HIV positive), indicated that specific components in the baseline peripheral blood flow could serve as predictors for a reduction in HVPG after administering direct-acting antiviral therapy in individuals with HCV-related cirrhosis [175]. The study by Yokoyama et al. [176] including 36 patients (12 patients with PH, 12 healthy controls, and 12 non-cirrhosis patients) showed a higher relative abundance of *Lactobacillales* (*p* = 0.045) and a reduction in *Clostridium* cluster IV (*p* = 0.014) (contains many butanoic-acid-producing strains, including *Ruminococcaceae* and *Faecalibacterium prausnitzii*) and cluster IX (*p* = 0.045) in patients with PH compared with other patients. Their results revealed no significant decrease in the *Bifidobacterium* genus in patients with cirrhosis. In the settings of this study, no distinction was made regarding the severity of PH or in comparison with cirrhotic patients without CSPH [177].

#### 3.3.3. Decompensated Cirrhosis/Acute-on-Chronic Liver Failure

ACLF is a syndrome consisting of the acute decompensation of CLD with the consequent development of multiorgan failure in the form of HE, hepatorenal syndrome, and circulatory failure [178]. It is characterized by a rapid progression of the clinical picture, poor outcome, and high incidence of mortality (in-hospital mortality rate of 45–65%) [179]. The triggers for the transition from the compensated to the decompensated phase of CLD are various and include infections, gastrointestinal bleeding, drug-induced liver injury, alcohol intake, portal vein thrombosis, and gut dysbiosis [63]. The development of ACLF involves the dysfunction of innate and adaptive immunities, and an important role is played by BT through the damaged intestinal barrier, inflammatory events, and immune disorders [180]. Alterations in the expression and function of pattern recognition receptors (PRRs) within intestinal cells, along with the detection of LPS by clustered TLR4 receptors on intestinal epithelial cells and immune cells (such as monocytes/macrophages and dendritic cells), can result in sustained activation and inflammation in the mucosa-associated lymphatic tissue (MALT). This process can trigger liver apoptosis and accelerate the progression of ACLF due to the presence of translocated LPS and other bacterial products, ultimately causing microcirculatory disorders in the liver [181,182]. Given that LPS is mostly removed via KCs, its dysfunction in ACLF leads to uncontrolled plasma LPS levels and endotoxemia resulting in uncontrolled SIBO and elevated levels of hepatotoxins, a chemical substance that further damages the liver [180]. A study by Chen et al. [183] investigated the fecal microbiota in ACLF patients and analyzed the temporal stability of the intestinal microbiota during the disease. The results of the study indicated a significantly lower microbiome diversity and richness in the ACLF group than in the control group. ACLF patients had a significantly (*p* < 0.01) higher abundance of *Pasteurellaceae* (mean 4.86% vs. 0.12%), *Streptococcaceae* (mean 4.66% vs. 0.34%), and *Enterecoccaceae* (mean 0.54% vs. 0.00%), while a lower abundance of *Bacteroidaceae* (mean 27.51% vs. 37.46%), *Ruminococcaceae* (mean 1.78% vs. 2.86%), and *Lanchnospiraceae* (mean 0.50% vs. 1.60%). The relative abundance of *Lanchnospiraceae*, a family including the genera *Butyrivibrio*, *Lachnospira*, and *Roseburia* known as butyrate producers, was significantly reduced in ACLF patients with HE compared to non-HE patients (mean 0.43% vs. 0.57%, *p* = 0.039), which may be explained by the suppression of these bacteria causing a decrease in SCFA production and thereby leading to an increase in colonic pH, ammonia production, and its intestinal absorption. The increasing abundance of *Pasteurellaceae* and MELD score were independent factors predicting the mortality rate in ACLF patients. In the study conducted by Solé et al. [28], patients with ACLF had a significantly lower richness of metagenomic species (MGS) among the different stages of cirrhosis, while patients with compensated cirrhosis had the highest richness. In addition, patients with ACLF exhibited an enrichment of MGS from *Enterococcus* and *Peptostreptococcus* species, coupled with a reduction in certain species like *Firmicutes* and *Roseburia*. Comparatively, patients with cirrhosis displayed an increase in *Bacteroides*, *Enterococcus*, and *Streptococcus* genera, while healthy individuals showed an increase in beneficial autochthonous bacteria such as *Eubacterium*, *Faecalibacterium*, and *Ruminococcus*. Furthermore, as the cirrhosis stage progressed, there was a significant rise in some pathogenic bacteria (*Enterococcus* and *Peptostreptococcus*) and a noteworthy decrease in several beneficial autochthonous bacteria (*Blautia*, *Dorea*, *Eubacterium*, *Faecalibacterium*, *Lachnoclostridium*, *Oscillibacter*, *Paraprevotella*, *Phascolarctobacterium*, *Roseburia*, and *Ruminococcus*). *Enterococcus faecium*, *Enterococcus faecalis*, and the MGS Homo sapiens correlated positively with the MELD score, while some species (*Clostridiales*, *Faecalibacterium*, or *Lachnoclostridium*) correlated negatively. A reduction in microbiome richness combined with the abundance of certain bacterial species (*E. faecium*, *S. thermophilus*, and *R. lactaris*) was associated with a high risk of short-term mortality.

Based on the results presented in the paragraph on the gut microbiome in different stages of CLD, most studies showed a tendency to increase the abundance of *Proteobacteria* and *Fusobacteria* and reduce *Bacteroidetes* in cACLD. In patients with PH, circulating nucleic acids of *Proteobacteria*, *Bacteroidetes*, *Actinobacteria*, and *Firmicutes* were the most abundant phyla in peripheral blood. At the same time, there have been no study results of microbiome profiles that could significantly predict PH severity yet. Research involving patients with ACLF demonstrated notable differences in their microbiota. Specifically, they exhibited a significantly lower richness of MGS along with a marked increase in particular pathogenic bacteria like *Enterococcus* and *Peptostreptococcus*. Additionally, there was a significant decrease in beneficial autochthonous bacteria, such as *Faecalibacterium* and *Ruminococcus*. A combination of these microbiome changes in patients with ACLF was associated with an increased mortality risk compared to controls. Also, the relative abundance of *Pasteurellaceae* and MELD score are independent predictive factors of mortality rate in ACLF patients.

It should be emphasized that, excluding cases of HE, a considerable portion of the mentioned research comprises mainly associative investigations. The definite causal relationship between the shifts observed in the microbiome during various CLD and stages of liver ailments remains somewhat uncertain. These alterations might either serve as causative elements driving liver conditions or might emerge as consequential adaptations to modified physiological states.

## 4. Modulation of the Intestinal Microbiome Composition and Its Effects on Liver Disease

### 4.1. Impact of Pharmacotherapy

Kalambokis et al. [184] demonstrated that rifaximin improves the hemodynamic state and renal function in patients with advanced cirrhosis by significantly reducing CO, increasing systemic vascular resistance, and decreasing plasma renin activity, levels of endotoxin, IL-6, and TNF-α. Sole et al. [28] observed that patients under chronic rifaximin treatment for the prevention of HE recurrence had significant changes in gut microbiome composition with an enrichment in eight MGS. Patients undergoing chronic norfloxacin treatment for SBP prevention and those receiving laxative treatment displayed distinct variations in their gut microbiome composition. Norfloxacin use in cirrhotic patients with increased levels of LPS-binding protein and hemodynamic derangement has shown positive benefits by decreasing endothelium NO-mediated vasodilation and BT [185]. According to Caraceni et al. [186], even minor alterations in the microbiome composition, including changes in Lactobacillus, Streptococcus, and Veillonella, induced by rifaximin, might be adequate to lower hyperammonemia and endotoxemia in cirrhosis. Additionally, the study emphasized that rifaximin led to an increment in cecal glutamine content and a decrease in the activity of small intestinal glutaminase, resulting in reduced ammonia production.

Furthermore, primary prophylaxis implementation using norfloxacin in patients with advanced cirrhosis and low protein ascites has been linked with a substantial reduction in the likelihood of developing SBP and hepatorenal syndrome within one year. Simultaneously, it has shown a noteworthy increase in the probability of survival at both the 3-month and 1-year marks [187].

NSBBs have a protective role against variceal hemorrhage in cirrhosis and SBP [51,52]. In cirrhotic patients, treatment with NSBBs has been associated with an increment in intestinal transit and a reduction in intestinal bacterial overgrowth, IP, and BT [53,54].

PPIs are known as a widespread medication used daily among numerous cirrhotic patients [47]. They have been associated with changes in the microbiota composition of cirrhotic and non-cirrhotic patients which could instigate SIBO or *C. difficile* infection [45]. Treatment with PPIs has been found to be a risk factor for HE and SBP in cirrhotic patients, with the risk increasing with the dose [47,48]. Yamamoto et al. [188] included 93 patients with CLD due to HBV, HCV, ALD, and NAFLD (62 PPI nonusers; 31 PPI users) and showed that CTP score, ascites, HE, and oesophageal varices were significantly higher in the PPI group than in the non-PPI group. Their study revealed that PPI usage in Japanese liver disease patients resulted in an increase in oral-origin microbial taxa and a decrease in autochthonous taxa. This alteration in the gut microbiota composition could pose a risk factor for HE or SBP. Therefore, it is necessary to prescribe this group of drugs judiciously to these patients.

### 4.2. Impact of Nutrition

SCFAs were associated with anti-inflammatory effects mediated by the activation of G-protein coupled receptor-43 [189]. Acetate supplementation has improved colitis, showing that SCFAs can reduce gut permeability and therefore hepatic toxicity [189].

Although the effect of a high-fat diet on the development of NAFLD and dysbiosis is already known, consumption of docosahexaenoic acid (DHA), omega-3 PUFA, presented a decrease in liver fat percentage in NAFLD patients [190]. The supplementation of omega-3 PUFAs demonstrated an induction of a reversibly increased abundance of *Bifidobacterium*, *Roseburia*, and *Lactobacillus* [191]. DHA supplementation demonstrated protection against acute ethanol-induced hepatic steatosis by inducing heme oxygenase-1, antioxidant stress, and hepatic cell survival and inhibiting stearoyl-CoA desaturase-1 and inflammatory cytokines [192,193,194]. Flaxseed oil rich in α-linolenic acid prevents liver damage by reducing endotoxin signaling mechanisms induced by TLR4 expression in KCs which recognizes CD14 causing activation of the MyD88 pathway and proinflammatory mediators (cytokines, free radicals) [195].

In one study, green tea consumption (500 mg tablet of green tea extract supplement per day) in patients with NAFLD was associated with a significant reduction in alanine and aspartate aminotransferase levels after 12 weeks [196], while another cross-sectional study did not find an association between green tea consumption (1–2 cups/day or ≥3 cups/day) and hepatic steatosis [197]. Ushiroda et al. [198] suggested that epigallocatechin-3-gallate (EGCG), the most abundant polyphenolic catechin in green tea, could alter BA metabolism and suppress fatty liver by improving the gut microbiota. In their study with high-fat-diet-fed mice, EGCG significantly induced a higher abundance of *Adlercreutzia*, *Akkermansia*, and *Allobaculum* and a lower abundance of *Desulfovibrionaceae*. Furthermore, EGCG significantly increased levels of serum primary BAs (CA and β-muricholic acid) and reduced levels of taurine-conjugated BAs (CA, DCA, β-muricholic acid). In the end, the researchers found that a correlation existed between the BA profiles and gut microbiota, highlighting the beneficial impact of Akkermansia and Desulfovibrionaceae in ameliorating BA dysregulation in mice fed a high-fat diet and treated with EGCG.

The consumption of three cups of instant coffee per day was associated with an increase in metabolic activity and/or number of *Bifidobacterium* which is considered to have beneficial effects on the gut microbiome and a preventive role in NAFLD development [199].

### 4.3. Fecal Microbiota Transplantation (FMT)

The administration of FMT yielded an increased gut microbiota diversity and beneficial taxa enrichment, leading to improved cognitive levels in recurrent HE patients. This improvement was notably superior to patients who received only the standard of care [200].

There have been reports on improved peripheral insulin sensitivity accompanied by alterations in plasma metabolites and intestinal microbiota in patients with metabolic syndrome who received FMT from lean donors. This improvement was observed after a period of 6 weeks and found to be significantly higher than that of the control patients who received a placebo [201,202].

Positive effects of the FMT have been observed in ALD patients leading to improvement in liver enzyme levels and better clinical outcomes [203]. Considering the very limited human data currently available, FMT in NAFLD patients showed a reduction in gut permeability, while not affecting hepatic steatosis or insulin sensitivity [204].

Considering a study with FMT showing an improved response to anti-PD-1 immunotherapy in melanoma patients [205], targeting the gut–liver axis represents a therapeutic option for HCC in the future [206].

### 4.4. Supplements and Probiotics

Lactic acid bacteria represent one of the most commonly used probiotics and many studies indicate that their use balances the microbiome composition in dysbiosis in ALD. *Levilactobacillus brevis* HY7410 and *Limosilactobacillus fermentum* MG590 were found to reduce blood alcohol concentration by boosting the activity of alcohol dehydrogenase and aldehyde dehydrogenase [207,208]. *L. brevis* MG5280 and MG5311, *L. fermentum* MG4237 and MG4294, and *L. reuteri* MG5458 demonstrated protective effects against HepG2 cell damage induced by ethanol. Furthermore, these strains were confirmed to be safe probiotics, as evidenced by antibiotic susceptibility and hemolysis assays [209].

Considering the above, they may be useful as new probiotic candidates for ALD prevention [209]. Studies have shown that *Limosilactobacillus reuteri* DSM17938, *L. brevis* SBC8803, and *L. fermentum* alleviate alcohol-induced liver damage in mouse models [210,211,212].

In a study which included 68 obese NAFLD patients, treatment with probiotics for 12 weeks resulted in a significant reduction in intrahepatic fat and body weight compared to the placebo group [213]. In a randomized, double-blind, placebo-controlled clinical trial, 52 patients with NAFLD were supplemented twice daily for 28 weeks with either a synbiotic which contained 200 million of seven bacterial strains (*Lactobacilluscasei*, *Lactobacillus rhamnosus*, *Streptococcus thermophilus*, *Bifidobacterium breve*, *Lactobacillusacidophilus*, *Bifidobacterium longum*, and *Lactobacillusbulgaricus*) and prebiotic (fructooligosaccharide) and probiotic cultures (magnesium stearate and a vegetable capsule) or a placebo capsule. The results of the study indicated a significant decrease in liver enzymes (AST, ALT, GGT), C-reactive protein, TNF-α, total nuclear factor κ-B, and fibrosis score as determined by transient elastography in the symbiotic group compared to the placebo one [214].

In the study conducted by Sharpton et al. [215], 21 randomized controlled trials were included, with 9 evaluating probiotics and 12 evaluating synbiotics. The treatment duration ranged from 8 to 28 weeks, and the trials involved patients with nonalcoholic fatty liver disease (NAFLD). Probiotic and synbiotic usage showed significant results, including a notable reduction in alanine aminotransferase activity and liver stiffness measurement (determined by elastography) and a significant improvement in hepatic steatosis (determined by ultrasound). Also, probiotics were associated with a significant reduction in BMI. Although promising, it should be noted there was a significant heterogeneity of the groups among the studies, and that none of the studies evaluated the histological response.

Oligofructose supplementation (OFS) in NASH patients significantly decreased alanine and aspartate aminotransferases after 8 weeks and insulin level after 4 weeks compared to the placebo [216]. Also, OFS showed improvement in liver steatosis and the overall nonalcoholic fatty liver activity score relating to the placebo. Additionally, OFS reduced LPS, IL-6, TNF-α, and Clostridium cluster XI and I, while enhancing *Bifidobacterium* spp. [217].

Lactulose is associated with a reduction in ammonia levels, a modification of the composition of the intestinal microbiota through an increase in beneficial bacteria (e.g., *Bifidobacteria* and *Lactobacillus*) and a decrease in potentially harmful bacteria (e.g., Enterobacteria), and an increase in the frequency and volume of bowel movements, which all together contribute to a healthier intestinal microbiota and better function of the intestinal barrier, reducing inflammation and improving cognitive functioning in patients with CLD [46,218].

The study by Grander et al. indicated that ethanol-induced intestinal *A. muciniphila* depletion could be restored by oral supplementation of *A. muciniphila*. Used therapeutically, *A. muciniphila* showed protection against ethanol-induced gut leakiness and mitigated hepatic injury and neutrophil infiltration [219].

In patients with AH who received probiotic therapy (cultured *Lactobacillus subtilis/Streptococcus faecium*) for 7 days, there was shown a significant albumin increase and reduction in TNF-α and LPS compared to the placebo [220].

Patients with cirrhosis who had recovered from an episode of HE were assigned randomly to groups given a VSL#3 probiotic or placebo daily for 6 months. VSL#3 contains four *Lactobacillus* species (*L. paracasei* DSM 24733, *L. plantarum* DSM 24730, *L. acidophilus* DSM 24735, and *L. delbrueckii* subspecies *bulgaricus* DSM 24734), three *Bifidobacterium* species (*B. longum* DSM 24736, *B. infantis* DSM 24737, and *B. breve* DSM 24732), and *Streptococcus thermophilus DSM 24731*. The probiotic preparation is composed of lyophilized bacteria in specific ratios, packaged as a granulated powder in each sachet, with a colony-forming unit concentration of 9 × 10^11^. In a group of patients who have received probiotic VSL#3, a lower number of hospitalized patients and complications of cirrhosis were observed compared to the placebo group. The probiotic group exhibited a hazard ratio for hospitalization of 0.52 (95% CI, 0.28–0.95; *p* = 0.034) compared to the placebo group, which carried a significant 48% reduction in the risk of hospitalization, therefore suggesting that treating five patients with the probiotic (VSL#3) for six months could prevent one hospitalization. In both the probiotic and placebo groups, the average time to hospitalization for any reason was found to be 136 days (95% CI, 122–150 days) and 109 days (95% CI, 93–124 days), respectively. A statistical analysis using a Chi-square test showed a significant difference between the two groups (χ2 = 4.93; *p* = 0.026). Child–Turcotte–Pugh (CTP) and MELD scores have been improved significantly over 6 months in the probiotic group [221].

Three months of probiotic administration (*Bifidobacterium breve*, *L. acidophilus*, *L. plantarum*, *L. paracasei*, *L. bulgarius*, and *Streptococcus thermophilus*) in patients with cirrhosis without overt HE significantly reduced levels of arterial ammonia, SIBO, and orocecal transit time, while it increased psychometric hepatic encephalopathy scores. Among the study participants, overt HE was observed in 7 subjects from the probiotic group and 14 from the control group (*p* < 0.05). The hazard ratio for the control group compared to the probiotic group was 2.1 (95% confidence interval, 1.31–6.53). Also, overt HE development was associated with psychometric hepatic encephalopathy scores, CTP scores, and SIBO [222].

The multistrain probiotic formulation (*S. thermophilus* DSM 24731, *B. longum* DSM 24736, *B. infantis* DSM 24737, *B. breve* DSM 24732, *L. paracasei* DSM 24733, *L. acidophilus* DSM 24735, *L. delbrueckii* subsp *bulgaricus* DSM 24734, *L. plantarum* DSM 24730) improved cognitive function and reduced the risk of falls in patients with cirrhosis and cognitive dysfunction and/or previous falls [223]. Furthermore, the aforementioned probiotic formula ameliorated the intestinal barrier and inflammatory response, showing a decrease in C-reactive protein, TNF-α, serum fatty acid–binding protein 6, and claudin-3 and an increase in a poststimulation neutrophil oxidative burst [223].

LGG was reported as a safe and well-tolerated probiotic associated with a reduction in endotoxemia, TNF-α, and dysbiosis in patients with cirrhosis [65].

Branched-chain amino acid oral supplementation in patients with CLD, cirrhosis, and HCC enhanced the activity of neutrophils and natural killer cells in the immune system and increased albumin serum levels, thereby reducing mortality [224,225].

The recent study by Philips et al. [226] indicated a predisposition to opportunistic infections in patients with decompensated cirrhosis, further worsening dysbiosis and therefore increasing the risk of sepsis, immunosuppression, and organ dysfunction. This further emphasizes the importance of identifying favorable supplements and strains which would have a positive effect on the immune system in patients with decompensated cirrhosis.

## 5. Unknowns and Future Goals of Microbiome Research in CLD

Despite the existence of numerous studies indicating the beneficial effects of various supplements (probiotics, prebiotics, synbiotics) in CLD, there remain many unknowns. For example, what is the mechanism by which a particular supplement improves CLD? Which supplement combination is the most effective one? What is the optimal duration of treatment? Therefore, we need further studies with larger sample sizes, longer follow-up, and more sophisticated omics-based diagnostic tools to assess the impact of different therapeutic protocols on the structure of the host microbiome, as well as to test these interventions against the liver histological categories and hard clinical end-points [22,227].

It remains to be discovered whether a significant gene and metagenomic richness reduction progresses from compensated to decompensated cirrhosis and its potential link to disease outcomes and pathogenicity in terms of disease complications and mortality.

That would require prospective studies based on ameliorating gut microbiome alterations and involving numerous patients [28]. Also, the confirmation of the hypothesis that marked abnormalities in the gut microbiome can affect the cirrhosis progression causing profound alterations in metabolism would require an evaluation of the main metabolic pathways [28].

Using advanced computational techniques and prosperous study designs, gut microbiome analysis combined with other clinical and diagnostic examinations represents a future standard for predicting disease sensitivity, defining CLD status, and providing personalized treatment through supplements, diet, and medication [160,228].

## Figures and Tables

**Figure 1 diagnostics-13-02960-f001:**
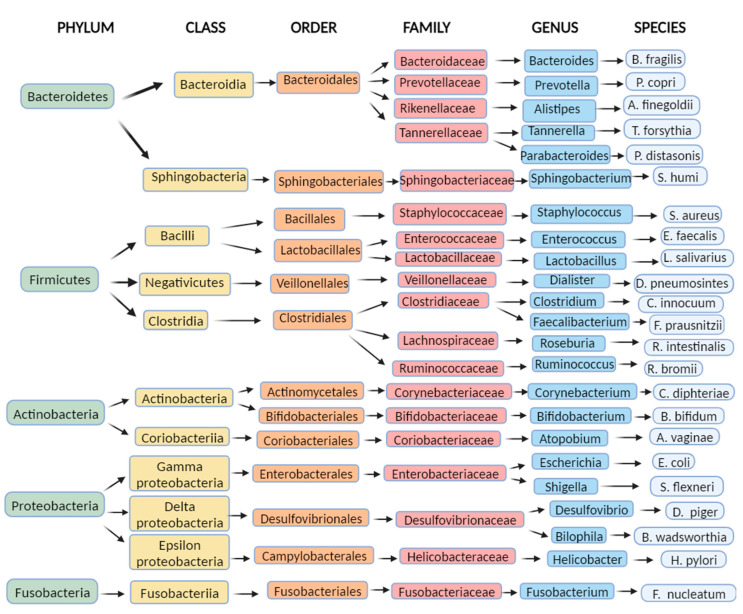
Examples of the most common bacteria in the human intestinal microbiome: a diagram representing examples of the most abundant bacteria of the gut microbiome by taxonomy (Created with BioRender.com accessed on 7 August 2023).

**Figure 2 diagnostics-13-02960-f002:**
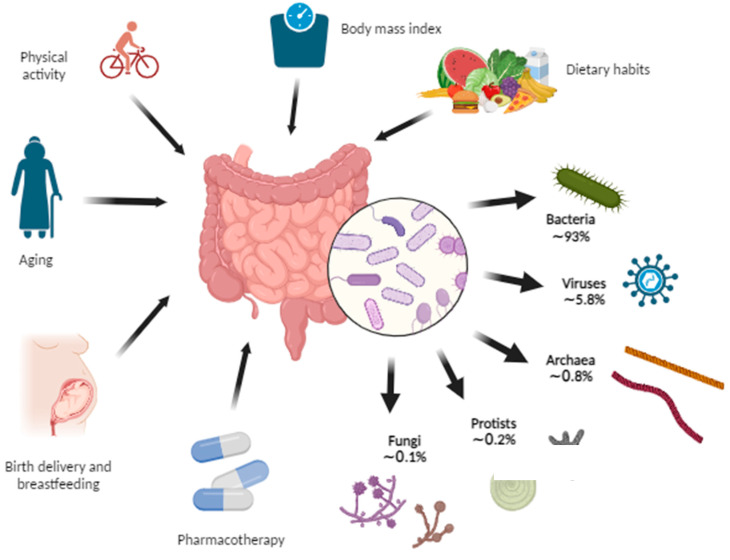
Gut microbiome composition and the influencing factors. The gut microbiome represents the collection of microorganisms including bacteria, viruses, archaea, protists, and fungi within the gut. Some of the factors that affect the gut microbiome are birth delivery, breastfeeding, aging, various pharmacotherapies, dietary habits, body mass index, and physical activity (Created with BioRender.com accessed on 7 August 2023).

**Figure 3 diagnostics-13-02960-f003:**
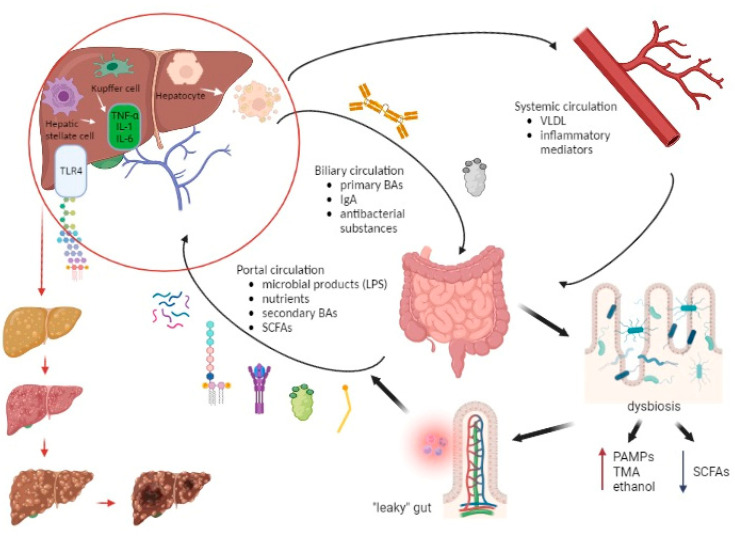
The interplay between the liver and gut microbiome along the development of chronic liver disease. The portal vein, biliary ducts, and enterohepatic recirculation represent the pathways through which the liver communicates with the gut. The portal vein transfers microbial products (LPS, lipopeptides, bacterial DNA, peptidoglycan), nutrients, SCFAs, and secondary BAs, while the biliary circulation delivers primary BAs, immunoglobulin A (IgA), and antibacterial substances from the liver to the gut. Also, the liver-derived metabolites (VLDL, inflammatory mediators) reach the bowels via the systemic circulation. Dysbiosis creates a predisposition for the formation of a “leaky” gut, which then leads to an increasing entry of pathogen-associated molecular patterns (PAMPs), trimethylamine (TMA), and ethanol into the portal bloodstream and a decreased entry of SCFAs. This results in an increasingly proinflammatory event in the liver in which LPS interacts particularly with TLR4 on KCs and HSCs resulting in the production of inflammatory cytokines (IL-1, IL-6, TNF-α). Also, LPS promotes lipogenesis and hepatocyte inflammation. Over time, these events in the liver lead to steatosis, fibrosis, and ultimately cirrhosis and the possible development of hepatocellular carcinoma. Abbreviations: BA—bile acid; DNA—deoxyribonucleic acid; HSC—hepatic stellate cell; IL—interleukin; IgA—immunoglobulin A; KC—Kupffer cell; LPS—lipopolysaccharide; PAMP—pathogen-associated molecular pattern; SCFA—short-chain fatty acid; TLR—Toll-like receptor; TMA—trimethylamine; TNF—tumor necrosis factor; VLDL—very-low-density lipoprotein. (Created with BioRender.com accessed on 7 August 2023).

## Data Availability

Data is contained within the article.

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
