# Peer review of "The Intestinal Microbiota in the Development of Chronic Liver Disease: Current Status"

_diagnostics, 2023, doi:10.3390/diagnostics13182960_

Round 1

Reviewer 1 Report

This is a very comprehensive review and it is quite lengthy, perhaps too lengthy as I needed a long time to read it.  One certainly wonders if it could be made more concise or could be split into two reviews (ie. part one and part two) as certainly the average clinical gastroenterologist and hepatologist will probably start reading it then give up.  With the exception of hepatic encephalopathy,  many of the studies cited tend to be association studies and it is not absolutely certain whether the changes found in the microbiome with various liver diseases and stages of liver disease are causal contributing factors or simply secondary changes to altered physiology.  The social determinants of gut microbiome in patients also needs to be mentioned as patients with end-stage liver tend to be in lower socioeconomic strata and therefore economically disadvantaged that must surely affect the microbiome (ie. food consumed because of cost, ability to safely prepare food, living environment including quality of drinking water etc).  These factors probably exist even in relatively affluent nations such as the United States.  

Author Response

Dear Reviewer 1,

We thank you for your comments, which we consider very constructive.

Related to the length of the work. We agree with you that the review paper is quite long, however, our intention was initially to cover most of the knowledge, including the most recent references in the field of intestinal microbiota and chronic liver diseases, so that people who are interested in the mentioned area could find most of the important information in one place. We believe that precisely this paper could enable the reader to introduce the issue itself, and then, through further reading, to open new horizons in thinking about the mutual influence of intestinal microbiota and chronic liver diseases.

Also, in the instructions for writing a review paper for the journal "Diagnostics", there is no limit on the maximum number of words, so we leave it up to the editors of the journal if they think that our review should be split into two papers. We believe that dividing this paper into two parts would lose the original idea, which is to provide the readership with comprehensive information in the field of microbiota and chronic liver diseases in one place.

As for the causality of the studies you mentioned, we agree with what was stated and we mentioned that part and added it to the paper at the end of paragraph 3.3.3. (marked with the accompanying comment "Added as recommended by Reviewer 1").

And finally, regarding sociodemographic determinants, we also added that part to the end of paragraph 2.2.4. and also in the subtitle of the paragraph itself (marked with the accompanying comment "Added as recommended by Reviewer 1").

Once again, we thank you for your comments and hope for your positive response.

Reviewer 2 Report

This narrative review is focused on the current knowledge about the composition of intestinal microbiota in healthy individuals and those with CLD, factors influencing microbiome composition, the impact of the microbiome on the liver and the mechanisms by which it occurs, possibilities and benefits of diagnosing microbiome composition in patients with different stages of CLD.

I suggest adding data about postbiotics in section: 2.2.8. Diet, probiotics, prebiotics

Also there are some different fonts, italics, different sizes of the fonts, i suggest correction.

Author Response

Dear Reviewer 2,

Thank you for your comments, which we consider very useful.

Regarding postbiotics, at the end of paragraph 2.2.8., a section about them was added and also in the subtitle of the same paragraph (marked with the accompanying comment "Added as recommended by Reviewer 2").

Likewise, throughout the entire text of the work, font sizes and styles were revised, as well as Italic markings in places where they were incorrect. Each change is marked with the accompanying comment "uniformized font size and type as recommended by the Reviewer 2".

Hoping for your positive response to our revised version of the paper.